# Debris-flow surges of a very active alpine torrent : a field database

Suzanne Lapillonne[1], Firmin Fontaine[1], Frédéric Liebault[1], Vincent Richefeu[2], and Guillaume Piton[1]

[1]Univ. Grenoble Alpes, INRAE, CNRS, IRD, Grenoble INP, IGE, Grenoble,France
[2]Univ. Grenoble Alpes, 3SR, Gières, France

**Correspondence:** Lapillonne Suzanne, (suzanne.lapillonne@inrae.fr)

**Abstract.** This paper presents a methodology to analyze debris flows focusing at the surge scale rather than the full scale of the debris-flow event, as well as its application to a French site. Providing bulk surge features like volume, peak discharge, front height, front velocity and Froude numbers allows for numerical and experimental debris-flow investigations to be designed with narrower physical ranges and thus, for deeper scientific questions to be explored. We suggest a method to access such
features at surge scale that can be applied to a wide variety of monitoring stations. Requirements for monitoring stations for the methodology to be applicable include (i) a flow height measurements, (ii) a cross section assumption and (iii) a velocity estimation. Raw data from three monitoring stations on the Réal torrent (drainage area: $2 \ \mathrm{km}^2$, South-East France) are used to illustrate an application on 34 surges measured from 2011 to 2020 on the three monitoring stations. Volumes of debris-flow surges on the Réal Torrent are typically sized at a few thousand cubic meters. Peak flow height of surges range from 1 to 2 m.
Peak discharge range around a few dozens cubic meters per second. Finally, we show that Froude numbers of such surges are near critical.

## 1 Introduction

The destructive nature of debris flows, as well as their sporadic behaviour, make debris-flow measurements in the field difficult. Monitoring of debris flow was pioneered in the 1970s (e.g., in Japan, Suwa et al., 2011) and more monitoring stations have
developed in the past 20 years (Hürlimann et al., 2019), allowing a wide range of debris-flow events in different torrent morphology to be observed. In their review, Hürlimann et al. (2019) show the various designs of the monitoring stations and their different objectives. debris-flow monitoring is performed for various purposes including understanding debris-flow initiation (Bel, 2017), increasing knowledge on the physics of the flows (Theule et al., 2017), and on impact forces (Nagl et al., 2022).

However, despite years of efforts in monitoring these phenomena, few data on debris flows had been shared in open databases. The collective effort and interest to gather such data would benefit from a structured method and definition of features of interest. One of the only available datasets was published by McArdell and Hirschberg (2020) who provided dates and bulk volumes of 75 debris-flow events measured on the Illgraben catchment in Switzerland. de Haas et al. (2022) published flow features (front height, velocity, flow rate, density, frontal shear stress), antecedent rainfall, and channel-bed elevation
change for the Illgraben torrent for 13 events.Marchi et al. (2021) also provided an extensive study on the Moscardo catchment (Italian Alps) presenting data on triggering rainfall, flow velocity, peak discharge and volume for the monitored hydrographs.

They made the complete dataset of debris-flow hydrographs and rainfall measurement for 26 events available in Marchi et al. (2020). In their paper, Comiti et al. (2014) published volumes, velocities, and dates of two events measured on the Gadria catchment in Italy as an initial analysis, with the same intent as the present work, namely to formalize and centralize data on debris-flow processes. Other events that occured on the same catchment were also described by Theule et al. (2017), Nagl et al. (2020) and in Coviello et al. (2021). Guo et al. (2020) made available velocities, flow depth, flow rate, flow width and duration of 23 surges on the Jiangjia Gully in China. Other data on debris-flow features can be found for the Chalk Cliff catchment in the United States (6 events by McCoy et al., 2012) and one event on the Cancia catchment in Italy (Simoni et al., 2020). These few interesting initiatives pave the way to community-driven open databases, they were however extracted from raw data with various approaches making difficult to pool them in a single consistent dataset.

Meanwhile, numerical methods improved tremendously in the recent years. Applications for debris-flow hazard mapping and design of mitigation measures are increasingly attracting attention, and allow always more scientific questions to be answered (Jakob and Hungr, 2005). These methods are now mature enough to model parts of the complex phenomena observed in the field at multiple scales. However, the lack of comparable, relevant, openly available, field data slows down the progresses in performing more realistic debris-flow modeling. This leads to a disparity between field reality and numerical and laboratory experiments. There is, for instance, a habit of exploring very large ranges of Froude numbers in numerical studies of impact forces, typically 1 - 8 (e.g., Albaba et al., 2015; Ceccato et al., 2018; Ng et al., 2020, among others). Performing such extensive parameter studies is a careful approach that ensure to cover the poorly known variability of Nature. However, it creates huge needs regarding experimental effort, computational power and time. These efforts are a high price to pay as they mean that more complicated scientific questions are not explored due to a lack of resources. In addition, in both experimental and numerical simulations, Froude numbers used are usually high, namely typically > 2 - 4 (e.g., Ng et al., 2020; Chen et al., 2020; Goodwin and Choi, 2022). Meanwhile, various regimes of impacts and flow behavior emerge depending on the Froude number (Faug et al., 2012), but the transition seem to occur for lower Froude values, typically near critical (Laigle and Labbe, 2017). Whether it makes sense to study each regime highlighted in laboratory experiments for field application should be decided in the light of field measurements. Thus, a database would ensure using features that are more representative of field reality, saving time to focus on deeper scientific questions.

Now that monitoring stations have been installed for a reasonable period of time, raw data processing is possible in order to build a common and open data base on flow characteristics of debris-flow surges. Such a database would aim to give access to the scientific community to values of typical flow features such as volume, maximal flow height, peak discharge and Froude numbers of real debris flows. A methodology for debris-flow surges data processing is described in the present paper to focus on the surge scale rather than full-scale debris-flow event (several fronts and surges with intermediate diluted flows). Representing accurately one debris-flow surge is already a great challenge to face for modelers, both numerical and experimental, and being able to have the physical feature of a surge will help achieving this challenge.

The end goal of this paper is to define a common methodology that is sufficiently simple to apply to make it widely usable to any automated debris-flow monitoring stations. Using it will then permit gathering characteristics of debris-flow surges in a homogeneous, easy to access database. Surge identification, velocity computation and volume determination methods are more

thoroughly described in this paper. The methodology we used to process monitoring data is first presented in this paper. Its application to the three monitoring stations of the Réal catchment in South-East France is then explained. The results describe the values of the surge parameters and show synthetically the interest of having several stations on the same channel in a
catchment. However, the methodology is not restricted to such monitoring scenarios. The ranges features of surges are first put into perspective with the literature. Potential relationships and evolution of surge features are then investigated and conclusive remarks are drawn.

## 2   Material and Methods

### 2.1   Methodology to compute the surge characteristics

#### 2.1.1   Concept of the event analysis

Each monitoring station has different types of sensors and different strategies to measure flow characteristics (Hürlimann et al., 2019). To apply the methodology, the following measurements are required (Fig. 1):

- flow height measurements with representative frequency sufficient to describe accurately the flow front rise on the hydrograph,

- known cross section where the flow is measured, *or* , an assumption on the relationship between flow height and wetted area. To reduce calculation errors, it is necessary to have a precise estimation of the wetted area before, during, and after a surge,

- a way to access directly the mean velocity of the surge, typically by estimating the travel time between a pair of sensors (eventually of different type) at sensible distance from one another, *or* , more accurate but rarely available, by direct
velocity measurement (e.g. image processing or large scale particle image velocimetry, see Theule et al., 2017).

These measurements must be done at sufficiently close locations to reasonably assume that the measured flow height is associated with the measured surge velocity. Between two sensors, there should be no major change in flow path, channel width and slope to ensure that the geomorphological processes are consistent along the interdistance.

The key parameters describing the surges are then computed using these time-series:

$$Q(t) = u \cdot A(t) \tag{1}$$

$$V = \sum Q(t) \cdot \delta t \tag{2}$$

$$Fr = \frac{u}{\sqrt{g \cdot h_{max}}} \tag{3}$$

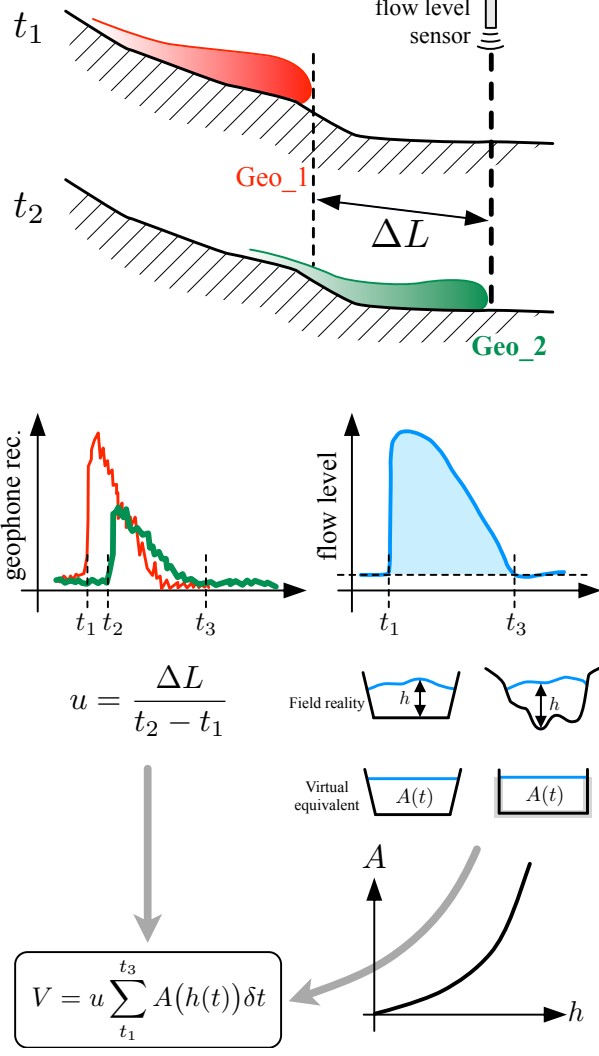

**Figure 1.** Synthetic overview of the method: a pair of sensor are used to estimate the time travel $\Delta t$ between known locations, and an assumption on the cross-section shape along with the flow depth sensor are used to computed the wetted area $A(t)$ and the associated surge parameters: discharge $Q(t)$, volume $V$ and Froude number $Fr$

where $Q$ is the debris-flow discharge [m³/s], $t$ is the time [s], $u$ is the mean surge velocity [m/s], $A$ is the wetted section [m²], $V$ is the surge volume [m³], $\delta t = \frac{1}{f}$ is the time sampling interval [s], $Fr$ is the Froude number [-], $g$ is the gravitational acceleration [m·s⁻²] and $h_{max}$ is the maximum value of the flow depth [m].

### 2.1.2 Surge identification

A debris flow is generally composed of one or several surges, with eventual intermediate flows that are more diluted (called "diluted runoff" hereafter) (Hungr, 2005). The strongest complexity, destructive power, and interest in debris flows is most probably the surges and their fronts. As a consequence, the database aims at gathering measurements focusing on the surge fronts and their main body, rather than the full scale of the debris-flow event including several surges ( e.g. as provided in McArdell and Hirschberg, 2020). In addition, it is arguable that diluted runoff have a lower sediment concentration and contribute much less significantly to the bulk event volume than the main, mature debris-flow surges. As a matter of fact, the applicability of Eqs. (1) and (2) rely on an assumption of high solid concentration (Hungr, 2005), constant throughout the surge. Focusing on data processing at the surge scale goes hand in hand with the intention for this database to be used to explore scientific question on the surge front behavior. This approach is different from other initiatives in the literature where the full scale of the event was considered.

Clearly defining the surges is thus a prerequisite to the data processing as the volume of the surge is integrated over the surge duration (Eq. 2), not the full event duration. If several surges in a single event are identified, each surge is taken separately as a data-point of the database.

The most basic identification of the surges is performed on the flow height time-series by identifying surges on the flow hydrograph. Doing so without cross control based on other information is however doubtful on catchments where diluted runoff and debris floods are frequent and intense. By experience, when available, images of the front can be used to define this separation. Geophones data proved to enable more reliable and data-driven criteria because they capture the solid transport intensity (Fontaine et al., 2017; Chmiel et al., 2022). Arattano et al. (2014) showed that the amplitude method for geophone signals allows to detect accurately the passage of a debris flow surge, while lightening data acquisition. Other methods, such as the impulse method, have shown accurate results for debris flow warning (Abancó et al., 2012). Bel (2017) showed that when mature debris flows travel at the levels of the geophones, this amplitude of the seismic activity is high and does not drop to zero. Immature debris-flow surge can also trigger instantaneously high geophone signal, but differ from mature debris flow because the signal frequently drops to zero during the event. This is why the criterion on determination between debris flows and immature debris flows cannot only be based on instantaneously high geophone signals. The existence of a prolonged period of consistently high seismic activity (with a high geophone signal) is chosen to differentiate debris-flow events from immature debris flows and debris floods. Diluted runoff are also easily differentiated from the surge using this method.

On Fig. 2, the concept of the identification is described. The onset of a surge is detected by both a sharp increase in flow height and a sharp increase in the amplitude of the geophone signal, followed by a consistently non-zero seismic activity. The end of a surge is either determined by a seismic activity dropping to zero or by the onset of a second surge that can clearly be separated from the first one. Indeed, at the end of the first surge of the figure, a drop in seismic activity is clearly observed and

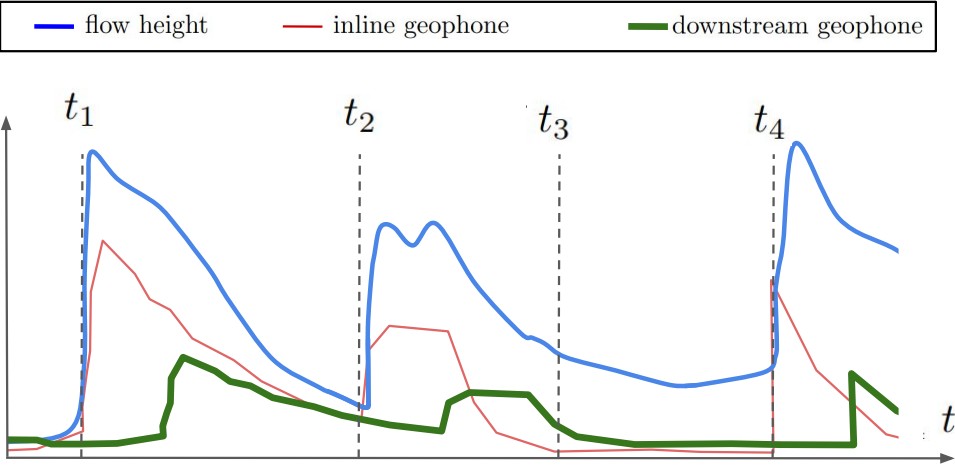

**Figure 2.** Conceptual graph explaining the surge identification approach: $t_1$ marks the onset of the first surge : sharp increase of energy in the geophone aligned with the flow sensor and sharp increase in directly measured flow height; $t_2$ marks the end of the first surge and the start of the second surge : geophone activity decreases before a sharp increase due to a second surge; $t_3$ marks the end of the second surge : seismic activity is negligible even though the flow height is still high: those are the diluted runoff flows, $t_4$ marks the start of the third surge. Note that even though the second surge has two peaks on the flow height, it is seen as one surge due to continuous seismic activity

a second sharp increase announces a second surge. On the other hand, the second surge displays two peaks in the flow level but as the seismic activity stays consistently high, those two peaks are considered part of one single surge.

### 2.1.3 Velocity calculation

In the proposed approach, as shown in Eq. (1), a single velocity value is considered for each surge. By doing so, the authors knowingly assume that the velocity is uniform within the surge. This is a crude simplification of the complex rheology of debris flows. The assumption is however required due to the lack of more precise data on most monitoring sites (see an exception in Nagl et al., 2020). This surge average velocity is a relevant proxy of the front velocity. Carefully defining the surge main body and consistently not including diluted runoff is a pivot point of this approach, as this approximation on the velocity is more relevant if the surge is only restricted to its front and main body (see section 2.1.2).

The velocity is generally computed using the lag $\Delta t$ between the signals of two sensors and the known inter-distance $\Delta L$ between those sensors. The distance is taken as the average flow path between the sensors i.e. the path of the main channel between the two sensors. Once the lag is determined, the velocity is computed as $u = \frac{\Delta L}{\Delta t}$. Accessing the value of this lag is done by comparing the two signals and their time-scale characteristics. Choosing two sensors that are at a sensible distance one from another is important: choosing two sensors too close to each other will induce significant uncertainty in the lag measurement. Due to the direct comparison of signals, the approach assumes that the source of the signal is the same that was propagating between the two different locations; in other words, the same surge is detected at both location. This approach thus

also assumes that the surge does not significantly change between the two sensors e.g., no massive deposition or erosion, no strong change in surge duration, no merging between surges. However, the travel distance should be sufficiently longer than the uncertainty on the lag to provide an accurate estimate. Two methods were used to estimate velocities : cross-correlation of signals if they were good enough and a visual identification method otherwise. For more information, the detailed methodology is presented in supplementary data.

### 2.1.4 Wetted area

From raw data, flow height and wetted area are determined at each time step. This requires assumptions on the channel bed level. Two examples will be presented in this section : assumptions that are reasonable on a check dam, and assumptions on a natural cross-section.

On controlled cross-sections, e.g., on a check dam crest, it is assumed that there is neither erosion, nor deposition. Conse-
quently, the bed level and cross section shape are assumed constant and known. Flow height and wetted area can then easily be estimated. This configuration is preferable. Practically this means $h_{effective} = z_{measured} - z_{dam}$, where $h_{effective}$ is the effective flow height [m], $z_{measured}$ is the level of the free surface measured by the sensor [m] and $z_{dam}$ is the check dam crest level [m]. The wetted area shape can be more accurately described by taking into account its convex surface shape (cross-section wise) (see Jacquemart et al., 2017).

Erosion and deposition occurring during debris-flow events may change the channel geometry. Not only does this mean that $h_{effective} \neq z_{measured} - z_{bed}$ where $z_{bed}$ would be the bed level before the flow [m], but it also means the cross-section shape will change during the event. The erosion-deposition process has two consequences : uncertainty on the channel shape and uncertainty on the channel bed level at a given time during the surge.

Accounting for the variability in channel is necessary (e.g. width, bed level, shape). Due to the debris-flow event, scouring
or filling can occur both vertically and horizontally to the cross-section. For each station, assumptions on cross section shape have to be made, and questions about variability in the channel have to be answered. For example, assumptions on cross section shape and change must answer to whether the channel can be scoured/ filled in that section and whether there is a difference in the preferred channel between low and high flows. Assumptions have to be as precise as possible using the information on the channel at this point (e.g. local obstructions to the flow are known, non erodible banks).

Bed level change throughout the surge is explored using different assumptions (Fig. 3 and as seen on 11). With $z_{low,min}$ the minimal bed level through the event, three assumptions are made, when relevant :

- The whole depth of the flow is sheared (effective ) until $z_{low,min}$ during the whole surge (**assumption max**),

- The flow isn't sheared in depth, this is less likely but allows to compute a minimal possible volume (**assumption min**),

- In the case of an erosion process, the bed level is assumed to follow a fitted logarithmic law following Kaitna and Hübl
(2021) (**assumption log**),

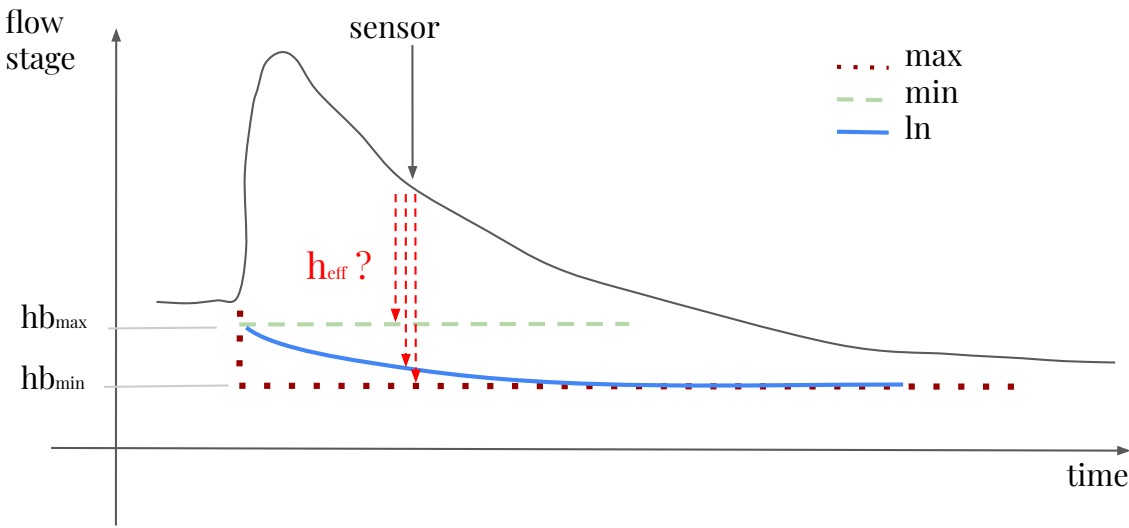

**Figure 3.** Assumptions on the bed level used to compute the efficient flow height in a natural cross-section: assumption **max** maximalizes the effective flow height, assumption **min** minimizes the effective flow height

## 2.2 Characteristics of the monitoring stations

The Réal Torrent, located in south of France, has been instrumented since September 2010 (Navratil et al., 2011). Three monitoring stations are distributed along the channel. Fig. 5 shows the station locations. The first one $S_1$ is located on a 20m wide check dam as seen on Figure 8a and is the most upstream. Station $S_2$ and $S_3$ are located in the middle reach and at the outlet of the torrent, and are both on natural cross-sections. In Table 1, a summary of the main physical features of the stations is shown (drawn from Bel et al., 2017). The purpose of the installation is to monitor the flow height, rainfall and seismic activity during sediment activity from bedload to debris flow. A thorough study of the station can be found in Fontaine et al. (2017) and in Bel (2017). The methodology presented above has been applied to these three stations and the results are presented further in this paper.

In essence, each station is equipped with : (i) a tipping bucket rain gauge with $0.201$mm resolution (Campbell), (ii) an ultrasonic or radar flow stage sensor (Paratronic), (iii) a set of three vertical geophones (GS20DX0 Geospace) each spaced out $\approx 100$m apart from each other, upstream, midstream and downstream of the flow height sensors.

Images of the channel and flow proved to be useful to facilitate the interpretation of the signals (Piton et al., 2017). Two cameras have been added to stations $S_1$ and $S_2$ (CC640 Campbell, replaced in 2018 by a PC900 Reconyx and EOS1200D Canon, respectively). Data are recorded using an environmental datalogger (CR1000 Campbell) powered by a solar panel, and are stored in a compact flash module (CFM100 Campbell).

**Table 1.** Physical features of the three monitoring stations

| Station ID | Elevation | Drainage area | Channel width | Channel slope | Type of section | Distance to downstream station |
|:---:|:---:|:---:|:---:|:---:|:---:|:---:|
| Units | (m a.s.l) | (km$^2$) | (m) | (m/m) | | (m) |
| $S_1$ | 1450 | 1.3 | 8 | 0.18 | Check dam | 757 |
| $S_2$ | 1340 | 1.7 | 7 | 0.14 | Natural | 908 |
| $S_3$ | 1254 | 2.0 | 12 | 0.11 | Natural | - |

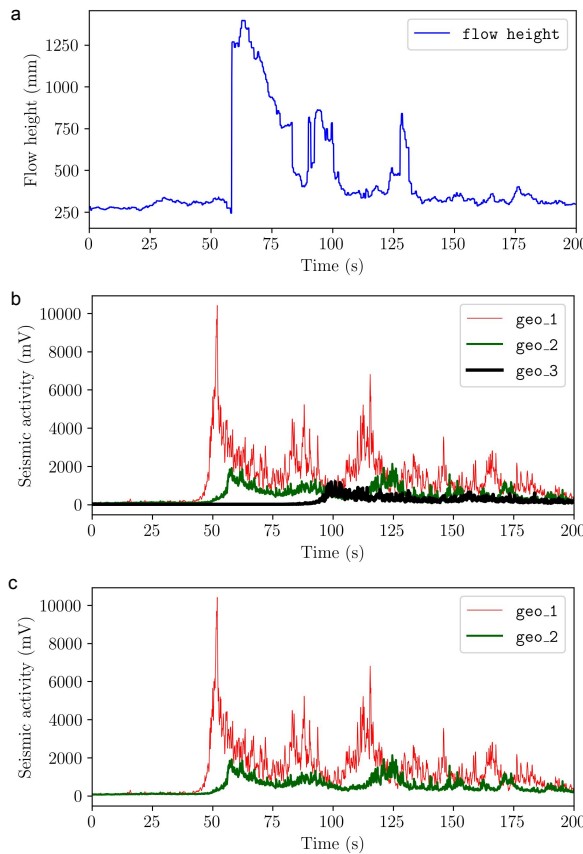

**Figure 4.** Overview of a recording of an event for station $S_1$ `geo_X` are geophone signals a) Flow height sensor b) Full record of the geophone signal c) Chosen signals

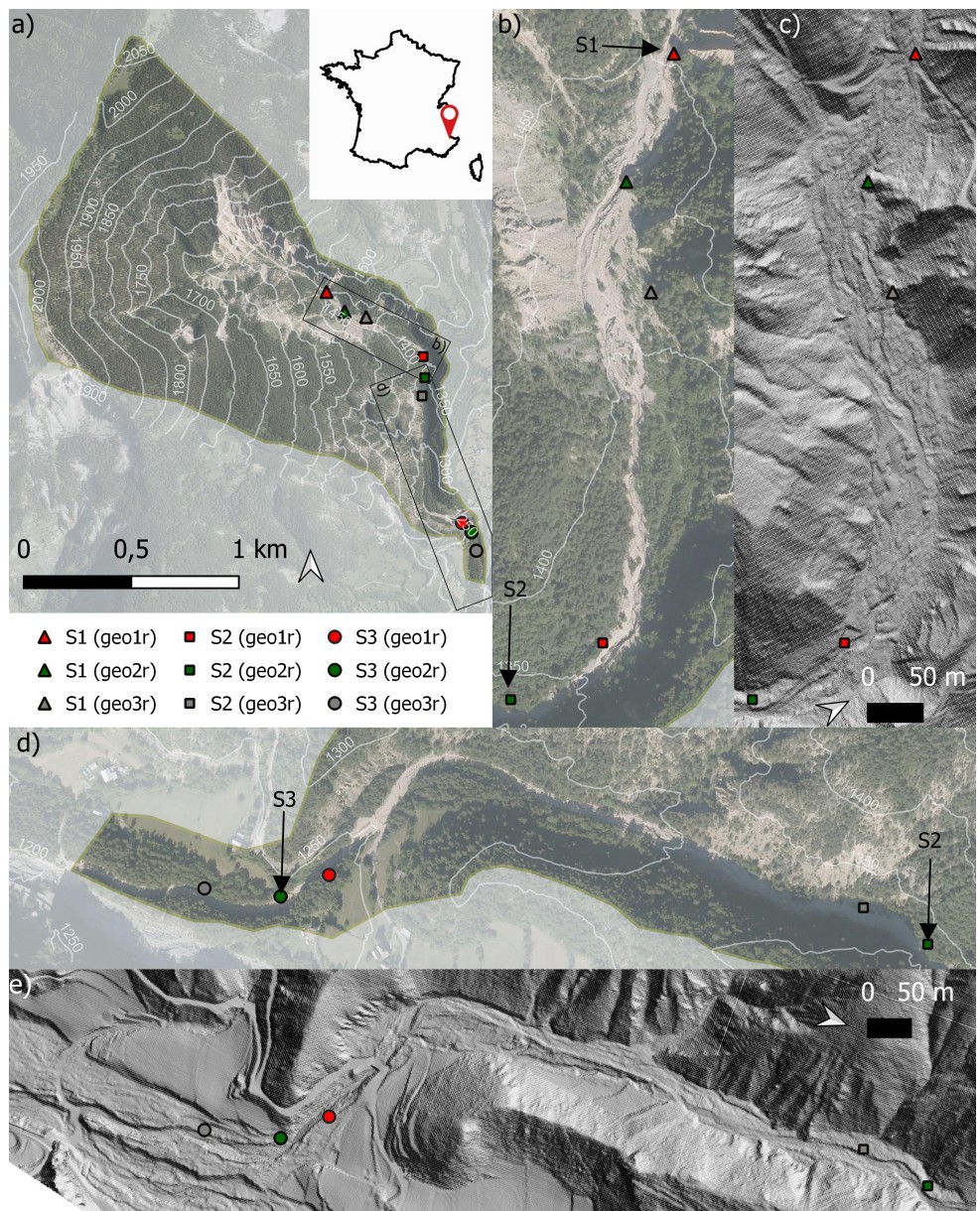

**Figure 5.** Overview of the installation on the Réal torrent : a) Full location of the torrent and its stations, drainage area is highlighted, and the three stations, arrows show the position of the flow height sensor, `geo_XX` denominates the geophones at each station (`r` or `l` signifies right or left bank) b) Station $S_1$ aerial photography c) Station $S_1$ Digital Elevation Model (D.E.M.) d) Station $S_2$ and $S_3$ aerial photography e) Station $S_2$ and $S_3$ D.E.M. (aerial pictures from BD ORTHO of the french geographical survey IGN)

On Fig. 4a, a complete set of measurements for one debris-flow event on station $S_1$ exemplifies the data analysis on one event. Out of these raw measurements, best suited signals are chosen by the user, as seen on Fig. 4b :

- For flow height along the event, if multiple flow height signals are available, the most reliable one is chosen, i.e. the flow height sensor that does not present any artefact (e.g. unphysical values, very noisy signal). Choosing consistently the same sensor across all events when it does not have any malfunctions is preferable. Here, only one is available,

- For the surge identification, one geophone signal is chosen, associated with the flow height signal. The sensors best suited for surge identification are those aligned with flow height sensors (see Fig 5: e.g. `geo_2`),

- For velocity determination, two geophone signals are chosen for cross-correlation. They must have the clear appearance of the debris-flow behaviour, with the continuously non-zero geophone signal explained in section 2.1.2, and be at a sensible distance one from each other: e.g. `geo_1` and `geo_2`.

The selection is mainly based on a visual estimation of which sensor is the most appropriate. The influence of that choice remains marginal.

This leads to Figure 4b with only the datasets used for the determination of the hydraulic values of interest. For each of these measurements, surges are identified and their features are computed. The user cross-controls the measurements and eventually goes for the visual method if the cross-correlation does not provide satisfying results (irrelevant value of velocity, low correlation coefficient or inconsistent velocity when compared to a first quick manual computation). The visual method consists in manually inputting the date of the onset of the surge on each geophone and considering the difference as the lag (see Fig. S3 in supplementary material). This visual method was used marginally, i.e. for one surge in our case, and was confirmed using image processing.

These sensors and post-processing allow to have for each event the followings : (i) seismic activity at three different points around the station with a frequency of 5 or 10Hz, (ii) rainfall data every 5mn (not used directly in this work), (iii) flow height with a frequency of 5 or 10Hz, and (iv) imagery of the event (when possible) with a 0.2 or 1Hz frequency,

Further in the paper, the value of the effective flow height is taken as :

- in the case of a controlled section, the mean value between the two assumptions for the section shape, as described in Bel (2017),

- in the case of erosion in a natural section, the logarithmic assumption,

- in the case of deposition in a natural section, the mean value between the min and max assumptions.

**Table 2.** Summary of the available data : black cells corresponds to available data, gray is non-applicable and crossed out cells are event that were detected but for which the data was not retrieved due to faulty sensors

| Date | Nb surge $S_1$ | $S_2$ | $S_3$ | Volume $S_1$ | $S_2$ | $S_3$ | Peak discharge $S_1$ | $S_2$ | $S_3$ | Froude number $S_1$ | $S_2$ | $S_3$ | Maximum height $S_1$ | $S_2$ | $S_3$ |
|---|---|---|---|---|---|---|---|---|---|---|---|---|---|---|---|
| 2011-06-29 | 1 | 1 | 1 | | | | | | | | | | | | |
| 2011-09-17 | 1 | 1 | | | | | | | | | | | | | |
| 2012-04-30 | 4 | 1 | | | | | | | | | | | | | |
| 2012-05-27 | 1 | | | | | | | | | | | | | | |
| 2013-03-30 | 1 | 1 | | | | | | | | | | | | | |
| 2013-05-18 | 3 | | | | | | | | | | | | | | |
| 2013-07-22 | 1 | | | | | | | | | | | | | | |
| 2014-01-04 | 1 | | | | | | | | | | | | | | |
| 2014-06-10 | 1 | | 1 | | | | | | | | | | | | |
| 2014-09-20 | 1 | 1 | | | x | | | x | | | x | | | x | |
| 2018-10-29 | 1 | | 1 | | | x | | | x | | | x | | | x |
| 2019-12-01 | 2 | | | | | | | | | | | | | | |
| 2019-12-19 | 4 | | | | | | | | | | | | | | |
| 2019-12-20 | 1 | | | | | | | | | | | | | | |
| 2019-12-21 | 1 | | | | | | | | | | | | | | |
| 2020-06-07 | 1 | | | | | | | | | | | | | | |
| 2020-06-13 | 1 | | | | | | | | | | | | | | |
| Total number of surges | 34 | | | 26 | 4 | 2 | | | | | | | | | |

# 3 Results

## 3.1 Observed debris-flow surges

For the construction of the database, only significant events were considered to ensure the analysis of mature debris flows: a threshold of flow height above $1\ m$ was selected for this catchment. This threshold is arbitrarily chosen from our experience on this particular catchments. Overall, $34$ events were considered for the Réal station for the period 2011-2020. Table 2 show when those events occurred, the number of surges passing at each station and the availability of the describing parameters. Over the 34 surges, most, i.e. 26, are recorded in the upstream station $S_1$, while only four surges reached $S_2$ and only two reached $S_3$, the most downstream station. The lack of events on the period 2014 - 2018 is partially due to the natural variability of event sizes but also due to faulty sensors during that time period.

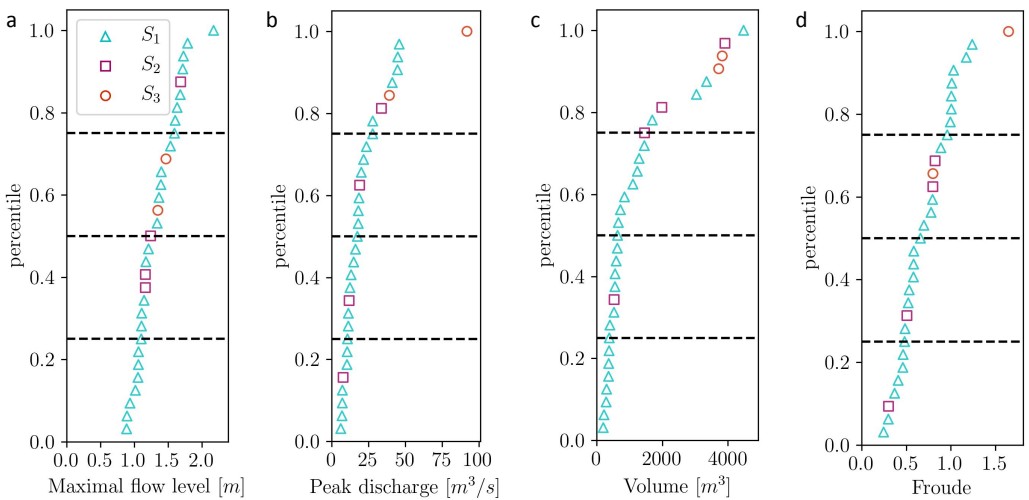

**Figure 6.** Cumulative density functions of hydraulic values of interest : a) Maximal flow level, b) Peak discharge, c) Volume, d) Froude

## 3.2 Distribution of surge parameters

One of the main interests of having an integrative dataset is to allow access to field ranges of hydraulic values of interest, such

as Froude numbers and volumes of surges. In Fig. 6, different cumulative distribution functions (CDF) of the data-sets are presented. Froude numbers range from 0.25 to 1.6, showing the range of regimes found in debris flows in our site. Whether this is a site specific feature or it can be shown on more sites that Froude number are typically critical would be a strong take home message for the community.

Surge volumes range from 200 to 4500 m$^3$ (Fig. 6c - quantile 25%, 50%, 75%: 390 m$^3$, 640 m$^3$ , 1460 m$^3$ ). Surges are

relatively small, typically from 1000 to 2000 m$^3$/km$^2$ (recall that this is surge scale and an event may comprise several of them, e.g., 1 - 4 in our observations of Table. 2, and some diluted runoff). Maximal flow height is most of the time lower than 2m (Fig. 6a - quantile 25%, 50%, 75%: 1.1m, 1.25m, 1.6 m). The peak discharge range between 6.2 and 91.8 m$^3$/s (Fig. 6b - quantile 25%, 50%, 75%: 10.8 m$^3$/s, 17.5 m$^3$/s , 27.9 m$^3$/s). The unit peak discharge is thus typically 0.775 to 7.65 m$^3$/s. Froude numbers range from 0.25 to 1.6 (Fig. 6d - quantile 25%, 50%, 75%: 0.48, 0.65, 0.95), i.e. are typically near critical.

The complete dataset is available in the supplementary data on Table S1.

Finally, relationships between these hydraulic values may be explored with a wider dataset, and a more thorough description of each event. Fig. 7 shows for instance the relationship between a few key variable (Froude numbers, volume of each surge normalized by the catchment area, front height and velocity). The surge volume was normalized by the catchment area to cross-compare measurements performed at different stations, but also to help transferring these results to other catchments.

A slight trend can be seen on Fig. 7a with increasing Froude number for increasing specific surge volume. While no clear conclusion can be drawn, there are no surges with large specific volumes ($> 1000 m^3/km^2$) which have clearly subcritical

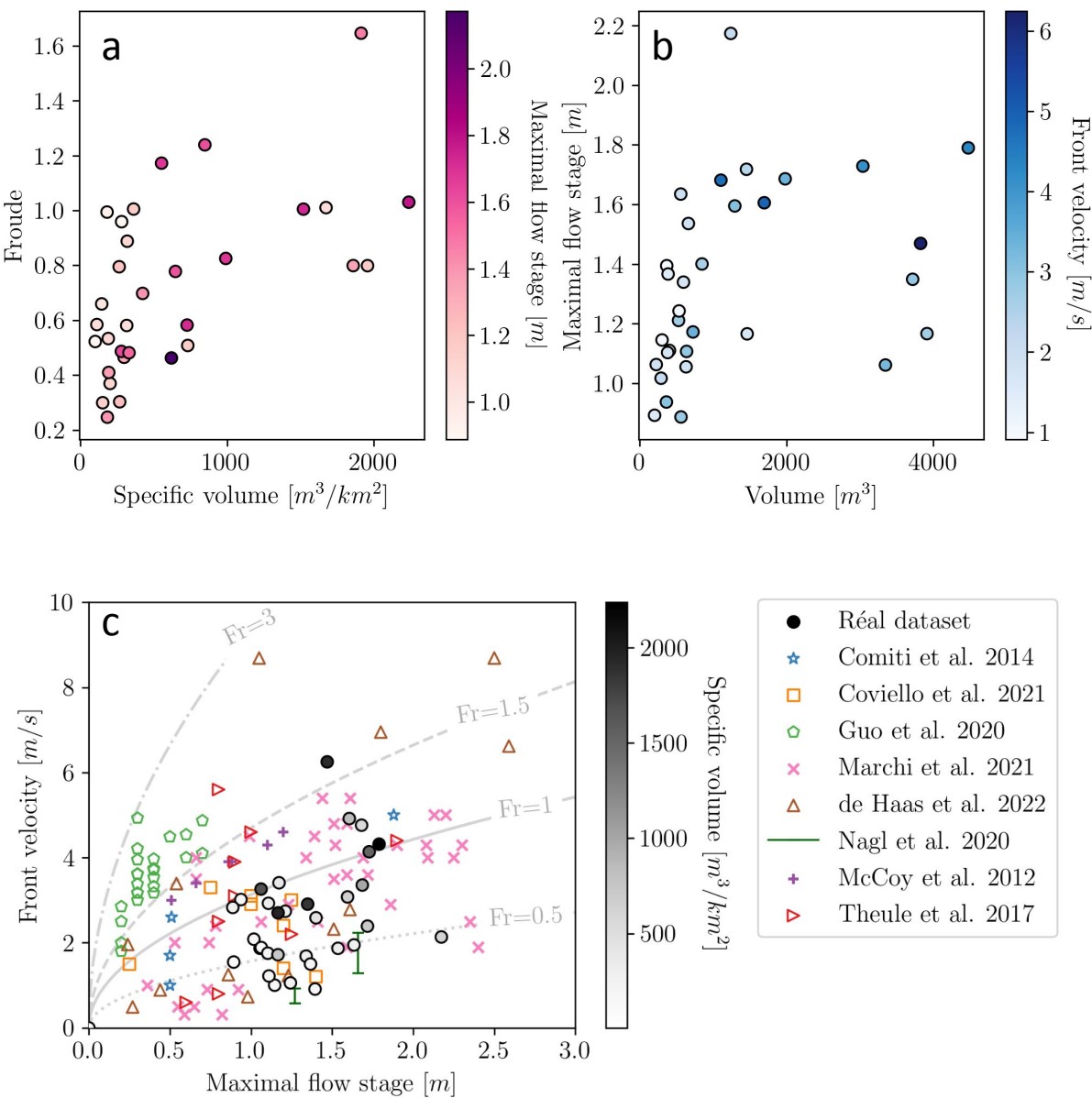

**Figure 7.** Examples of different relationships that can be explored with this dataset: a) Froude number VS specific surge volume, b) Maximum flow height VS surge volume and c) Front velocity VS maximum methodology. Data from the literature (Comiti et al., 2014; Coviello et al., 2021; Guo et al., 2020; Marchi et al., 2020; de Haas et al., 2022; Nagl et al., 2020; McCoy et al., 2012; Theule et al., 2017) is displayed on c) to contextualize the values: For Nagl et al. 2020 ranges of maximal and minimal values were taken. For Comiti et al. (2014) and Coviello et al. (2021) values of the flow height were estimated graphically. For Marchi et al. (2020), effective flow height was computed as the difference between flow height at the peak and the start of each surge. Colormapping is only showed for the Réal dataset. Grey lines display different Froude number relationships.

Froude numbers (all Froude numbers are above 0.8). Most of these surges have near critical Froude numbers. It seems that debris-flow surges of large volume require a strong inertial input to flow, as there are no subcritical Froude numbers for volumes of the selected range. Their heavy granular content, increasing their macroscopic viscosity,cause that subcritical, slower flows, with high volumes would stop or deconstruct. On the other hand, smaller surges can flow more easily and do not need strong inertial inputs to maintain steady flow. The fact that most of these surges are near critical might in part be due to the sampling at the stations and not the possibility for them to exist : very fast surges with high volume and high inertia are very rare in this catchment. Indeed, the hydrology of the catchment allows for sediment transfers to occur rather often (see Bel, 2017) and the moraine material and steep slopes lead to low yield criterion of the accumulated sediments. This means that the surges with high volume that are passing at the stations meet the "minimum requirements" to flow. One surge with supercritical Froude number and high volume is still detected.

If surges would all be of the same hydrograph shape and mixture composition, surge volume would be highly correlated with flow height. However, maximum flow height is quite variable with surge volume (Fig. 7b). This supports the argument that debris flow hydrographs vary widely.

Similarly, no clear correlation seems to appear between front velocity and flow height (Fig. 7c). Literature data has been displayed, drawing from Comiti et al. (2014); Coviello et al. (2021); Guo et al. (2020); Marchi et al. (2020); de Haas et al. (2022); Nagl et al. (2020); McCoy et al. (2012); Theule et al. (2017).

Our dataset ranges in similar Froude numbers as the literature, with most points between $Fr = 0.5$ and $Fr = 1.5$. A point from Simoni et al. (2020) would plot out of the figure (maximal values of velocity: $4m/s$, flow depth: $4.5m$, rendering a subcritical Froude number $Fr = 0.6$). Two point from Marchi et al. (2020) dataset have similar features, notably Froude number close from 0.6 and would also plot out of the figure. Most dataset show similar values than the Réal torrent with the notable exception of the dataset provided by Guo et al. (2020) that has generally higher Froude numbers. This is attributed to specificities of this catchment which do not have the slow laminar features that can be found on reach like the Réal torrent. Overall, all Froude numbers displayed stay under $Fr = 3$.

We interpret these lack of clear trend or correlation as evidences of varying surge mixture composition between events. The sample size remains however relatively small and site-specific, calling for careful interpretation of these data. We believe it will be of high interest if several other sites could be added to a similar analysis. Fitting a relationship between Froude numbers and surge volume could be a very interesting asset for numerical and experimental modeling.

## 4   Discussion

### 4.1   Relationship between surge parameters

Figure 6 and 7 show the ranges of the different features in the database for the Réal torrent. Specific volumes range from $101m^3/km^2$ to $2237m^3/km^2$. In comparison to specific volumes given by McArdell and Hirschberg (2020), which range from $171m^3/km^2$ to $7690m^3/km^2$ (catchment size $11.69km^2$), these are much smaller. One of the key reason why there is such a difference – apart from differences in geological and rheological makeup – is the method employed : classically, available

volumes can contain multiple surges and diluted tails and thus, volumes are not as restrictive as in the method employed in this paper. Specific volumes of the Réal catchment being much smaller is consistent with the difference in hypothesis in each methods. In Coviello et al. (2021), the Gadria catchment monitoring is described and the method employed is much more comparable. In that case, specific volumes of surges range from $35m^3/km^2$ to $952m^3/km^2$, when taking the catchment size as $6.3km^2$, which are of similar range compared to our dataset.

For smaller specific volumes ($< 1000m^3/km^2$), Froude numbers range from 0.2 to 1.2 with most surges being clearly subcritical with Froude $< 0.8$. Flow conditions for smaller volumes require less inertial input. For a same specific volume, a wide range of subcritical Froude numbers are found, showing that volume is not the main driver to flowing conditions, and that surge mixture composition varies widely in surges of low volume, i.e. $< 1000m^3/km^2$. This composition of the mixture changes the mobility of surges.

The initial expectation for Figure 7b would be that surges of higher volume render higher maximal flow height. This would be the case if hydrograph shape was consistent on all events. Debris flows have very variable flow hydrographs (Mitchell et al., 2022, among others) due to a wide range of flow mixture. This leads to similar volumes of debris-flow surges to be caused by different types of flow hydrographs : shallow surge which last for a long duration or very intense, high, but short surges.

Figure 7c shows no definitive relationship between proxies of inertial and potential inputs in the flow. This is yet another
argument to point out that surge granular content and mixture composition might differ widely from one event to another on the same catchment. The idea that composition of the debris-flow surges changes between events is supported by Hürlimann et al. (2003). A study of the surge content in boulders and coarse grain (Takahashi, 2014) and of their interstitial fluid rheology (Bardou et al., 2003) would be complementary to support this idea, but is at the moment not possible with the available data.

### 4.2   Evidence of the erosion/deposition cycles

On Fig. 5b and d, the valley bottom landforms bear the footprint of high morphological activity due to debris flows. More specifically in the reach between $S_1$ and $S_2$ where landforms such as abandonned channels, levees and lobes can be seen (Fig. 5b-c). Fig. 8 exemplifies these changes in the channel morphology directly downstream of station $S_1$ at five different dates. An erosion/deposition cycle of the channel incising and refilling is highlighted over six years of field pictures. Such processes explain why many debris flows are measured at station $S_1$ while much less are observed further downstream.

In Figure 9, volumes of all events are shown along time. If the geomorphic cycle exemplified in Figure 8 was detectable by this method, pseudo-cycles of cumulated volumes surges at station $S_1$ would be less frequently exported as surges of higher volume at station $S_2$ (or as many small volume surges at $S_2$ in the following years) i.e. if it were possible to see this geomorphic cycle, the cumulated volumes of the surges passing at S1 would be found to be equal to the cumulated volume at S2 over the years. Any of the deposit at S1 or between S1 and S2 would then be exported downstream. It can be seen that the
two surges reaching station $S_3$ are indeed of relatively high volume but the data lacking between 2015 and 2019 prevent us to draw further observations. With the current data, we can simply conclude that higher volumes of debris flows pass station $S_1$ than further downstream. The system is thus either or both storing sediment in the valley through aggradation and/or also exporting sediment volume through another process than mature debris flows. This is in agreement with the analysis in Theule

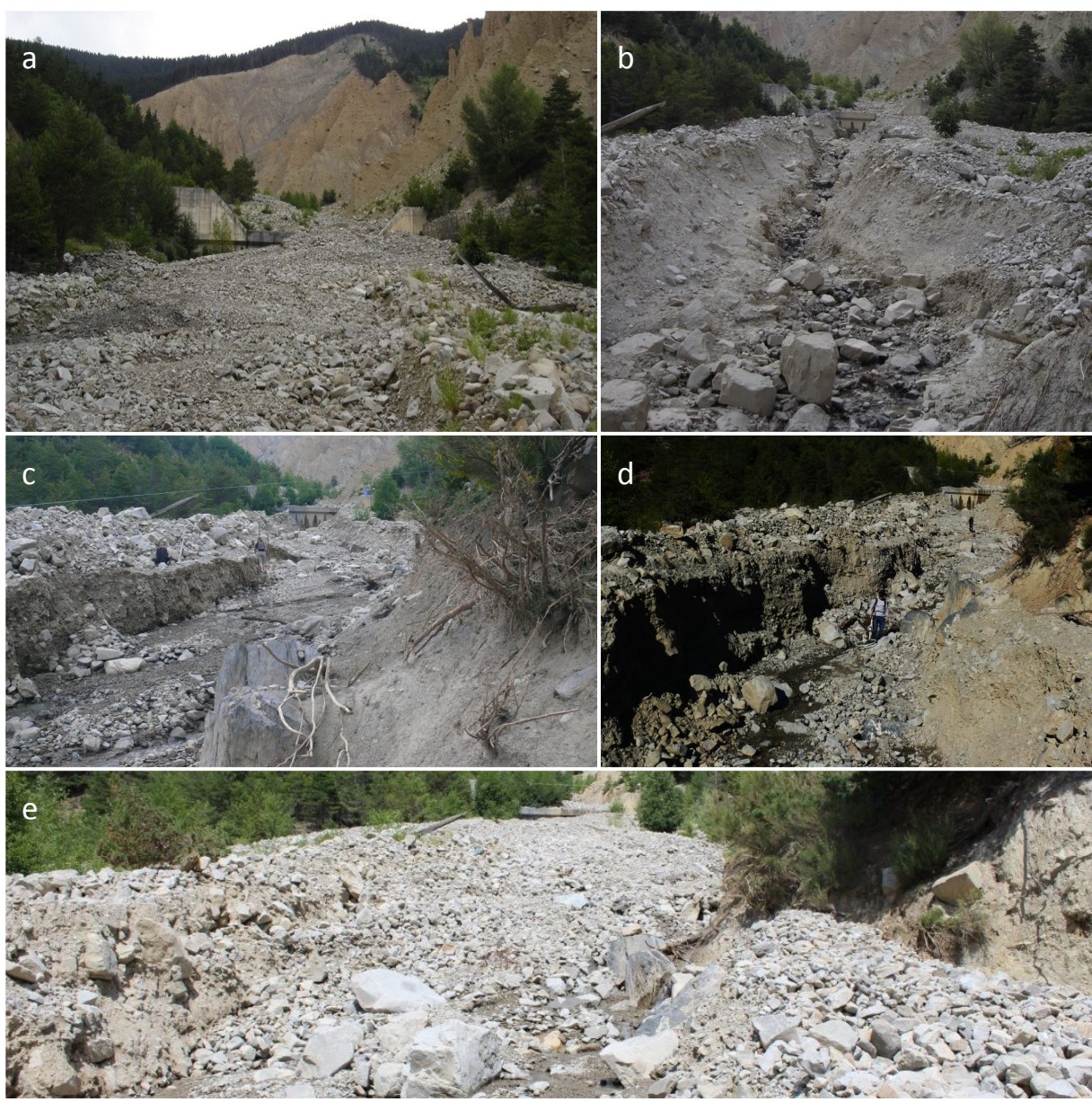

**Figure 8.** Pictures (from G. Piton) taken on the $S_1$ stations over 6 years a) channel filled in June 2009, b) channel deeply incised in July 2011, c) channel widened and partially refilled in June 2014 (person for scale), d) channel further incised October 2014 (person for scale), e) channel refilled in July 2014 (pictures form the authors)

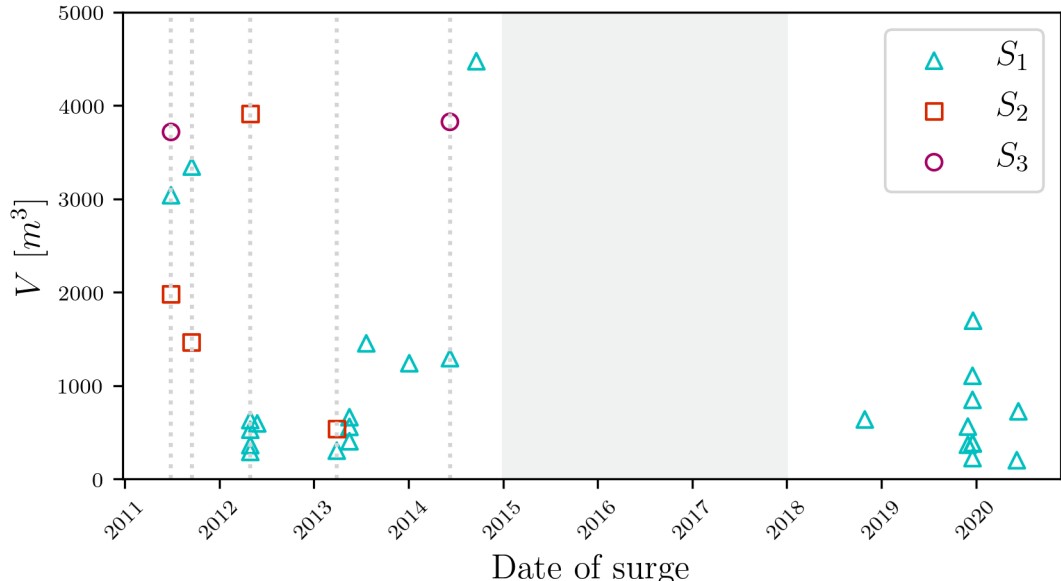

**Figure 9.** Volume of the surges of mature debris flow passing the stations, grey area has no data partly due to a faulty sensor invalidating measurements from 2016 until the end of 2017 when the sensor was replaced. No surges were detected in 2015. Grey dotted lines represent dates for which the surge was detected at multiple stations.

et al. (2015) which concludes that the sediment activity can be of transfer, erosion or deposition in these positions in the reach and in this range of slope ($0.11 - 0.18 m/m$, see Table 1). The applicability of this approach to study the sediment cascade is limited by multiple aspects: the first being that the data of interest is kept at the surge scale and focus on mature debris flows (threshold height > 1 m). Due to the way the data has been processed, studies on global sediment balance are not possible with this analysis, as the events of bed-load and wash-load are not taken into account. Indeed, despite its high debris-flow activity, the Réal Torrent experience other processes causing long term morphological changes as bed-load transport and debris flood that have meaningful impact on morphological changes and sediment fluxes in various parts of the catchment (Theule et al., 2012).

### 4.3 Upstream-downstream transfers of debris-flow surges along the channel

A key interest of having three different monitoring sub-stations on the same torrent is the possibility to study cascading sediment transfers. Fig. 10 shows the analysis of volumes, flow rates, Froude numbers and flow height of each events that could be found on more than one of the station. One could expect to see consistent relationships between upstream and downstream characteristics but results are more complicated.

Volumes passing stations $S_1$, $S_2$ and $S_3$ are generally very different at a same date (Fig. 10). In some cases, the debris-flow surges were growing, recruiting sediment from the bed ($V_2 > V_1$ and / or $V_3 > V_2$) showing the profound morphological

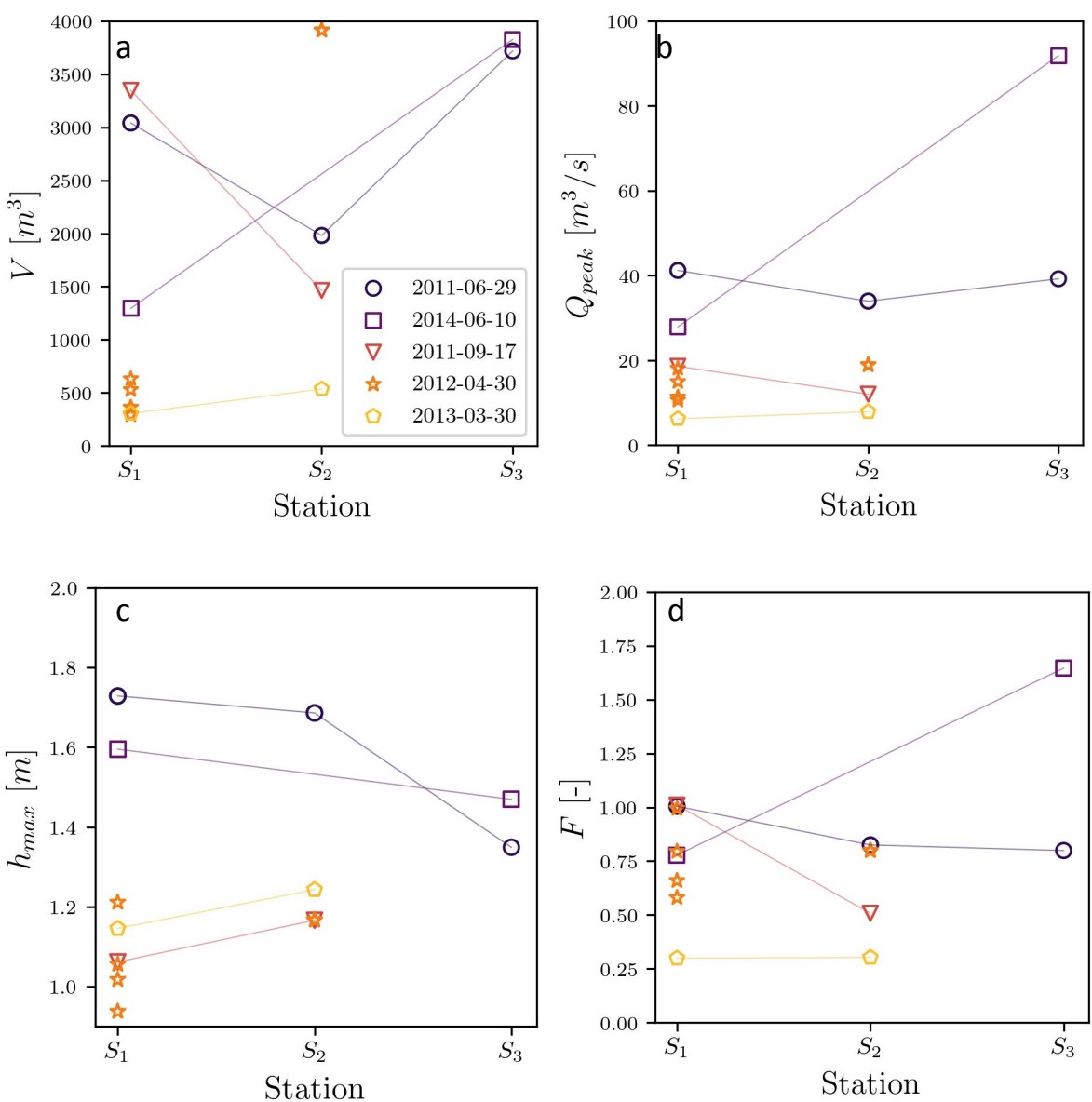

**Figure 10.** Temporal study for surges detected at two different sub-stations a) Peak discharge over traveled distance (from the beginning of the channel), b) Volume over traveled distance c) Maximum flow level and d) Froude number

changes debris-flow passage can lead to. In other cases, some deposition occurred ($V_2 < V_1$) but erosion might still appear
downstream. For the subset of events happening on the same date at the three stations, no particular relationship between the
four parameters studied in Fig. 7 was identified.

On Fig. 10a and b, volumes and peak discharge should consistently grow if the surges were consistently eroding from up-
stream to downstream of the reach. Events like the 2012-04-30 surges show increasing volumes, with a potential agglomeration
of the surges between $S_1$ and $S_2$ (accumulated volumes at $S_1$ are smaller than the volume at $S_2$). This shows deep erosion is
possible between the two stations, which is consistent with the morphological changes shown on Figure 5b. Nonetheless, on
this event, peak discharge is not increasing between the two stations. This specificity points out how he pure measurement data
and analysis benefit from more specific event data and description.

Similarly, maximum surge depth can also either be lower upstream (2013-03-30 of Fig. 10c) or higher at the first station
(events of summers 2011 and 2014, Fig. 10c). The Froude number also varies from upstream to downstream with some events
having lower downstream Froude number and others not (Fig. 10d). Froude numbers could be expected to be consistent from
upstream to downstream : the ability to flow of the surge would be driven by the interplay between kinetic and potential inputs.
Erosion and deposition processes of the surge along the reach will influence the Froude number both by changing the volume
and the composition of the surge. This is in agreement with the fact that the slopes in this section are in a sediment transfer
regime, as stated by Theule et al. (2015)

The observation on volumes, discharges and surge heights, as well as the much stronger frequency of mature debris flow
passing $S_1$ against those passing $S_2$ or $S_3$ (26, 4 and 2, respectively), highlight that strong processes of erosion and deposition
occur in the catchment.

While analysing data from three different stations located on such a small and active catchment is interesting, events detected
on multiple stations are scarce : most surges detected upstream tend to deposit or to attenuate while travelling such that they
are not detected as a mature surge downstream. On the opposite end of this spectrum, a surge that was under the detection
threshold on the upstream station might have become fully formed in the downstream stations (see the events of June 10, 2014
and October 28, 2018 that were detected at $S_1$, not at $S_2$ and again detected at $S_3$, Tab. 1).

On the other hand, surges that are detected on multiple stations are also difficult to rely to each other, and although volume
comparison could be interesting, actual quantitative comparison relies on the hypothesis that the exact same surge between
upstream and downstream stations is comparable, i.e. that along the journey, only marginal changes in process occurred,
which is known to be a crude hypothesis of this first work. In essence, the data shown in this paper are interesting because
they are actual field observations with quantitative measurements but the analysis of the catchment sediment transfers is not
possible. However, the dataset does demonstrate how strong and intense the processes of erosion and deposition in debris flow
prone catchments are. An analysis seeking to determine rainfall triggering conditions of debris flows would for instance draw
different conclusions depending on which station is used (but see Bel et al., 2017, which partially addresses this issue). We
believe that further effort should be put on better understanding not only debris-flow triggering factor but also propagation
through headwaters and intermediate reaches.

Additional multitemporal high-resolution images would help drawing conclusions on this temporal investigation, and such field campaigns would help answer some of the remaining questions such as remobilization of the deposited material, evidence of the global pseudo-cycles, etc (e.g. Cucchiaro et al., 2018, 2019a, b).

### 4.4 Analysis of the ranges of the physical characteristics of the events

Comparing the present data to the literature shows the ranges of volumes and flow rates found in the Réal torrent to be consistent with empirical fits proposed in previous works (Bovis and Jakob, 1999; Rickenmann, 1999; Mizuyama et al., 1992), even though the measurements of volumes were done with debris-flow levees in these previous works rather than direct measurements as our contribution. More precisely, these fits are using the full-scale debris flow event rather than a single debris flow surge. On Fig. 11, three values are always plotted for the Réal database : they compare the maximizing, the minimizing and the value of the wetted area chosen to be saved into the database. The effect of the choice of assumption stays relatively marginal for the upstream station but does have a significant effect for the natural corss-sections, as expected. This highlights the importance of these assumtions on the processing of raw data.

According to Fig. 11, the peak discharge of the Réal catchment for various volumes of debris-flow surges seems closer from the empirical fit related to granular debris flows of Bovis and Jakob (1999) or the fit proposed by Rickenmann (1999). Peak discharges associated with muddy debris flows are lower than those measured at the Réal catchment for equivalent volumes. These results are consistent with the work of Bel (2017) who already showed this concordance using an analysis considering the full debris-flow event with a former version of this methodology.

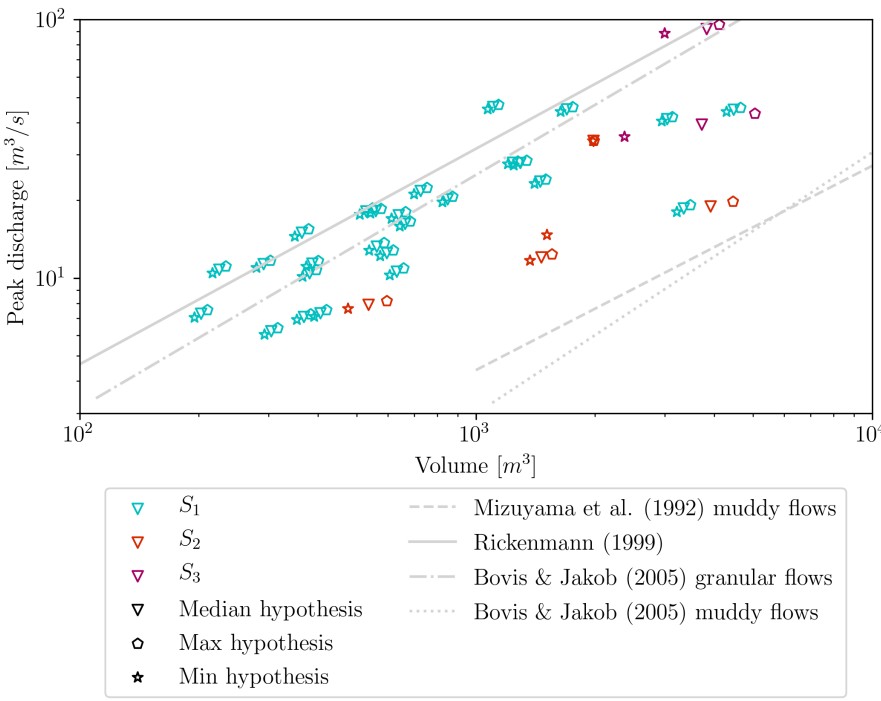

**Figure 11.** Relationship between debris-flow surge volume and peak discharge for all three stations of the Réal torrent (color scale for the station and dot shape for the assumptions on the bed level) - Comparison with empirical fits of datasets from the literature (Bovis and Jakob, 1999; Rickenmann, 1999; Mizuyama et al., 1992)

# 5 Conclusions

This work is a conceptualization of a widely applicable methodology for debris-flow surges data processing from monitoring stations. A full and simple methodology on debris-flow data processing is presented. The clear goal of this paper is not only to make a first dataset for the Réal torrent using this methodology available but also to call for collaboration on a common database for debris-flow surge features.

Bulk surge features are investigated including volume, front height, peak discharge and Froude number. This investigation allowed to access these hydraulic features on 34 surges gathered from 2011 to 2020 on the Réal torrent catchment (South-East France, catchment size 1.3 - 2 km$^2$). Surge volumes are typically a few thousand cubic meters, peak flow heights range from one to two meters, peak discharge is usually of the order of magnitude of a few dozens of cubic metres per second and their Froude number is near critical.

Access to representative field data will ensure accurate representation of these natural flows. This database is meant to be extended to other monitoring stations to strongly gain in impact on the scientific community Open access to field data for numerical research can be the bridge needed to close any gaps between the field-driven approaches and the numerical investigations. Research on debris-flow behaviour is growing and we hope that this initiative will allow more projects to be born, and allow field observations and numerical computations to evolve conjointly. On top of this, experiences drawn from the post processing of such data can allow for better, more effective data monitoring in the future (e.g. what type of cross section to choose, where to install successive stations).

*Data availability.* The processed data is available in the supplementary data of this paper. The raw data (geophone signals, flow sensors and rain accumulation) are available upon reasonable requests to the authors

*Author contributions.* Conceptualization: S.L. and G.P., data curation: S.L. and F.F., Methodology: S.L., F.F. and G.P., Supervision: F.L. and G.P., Visualization: S.L., V.R. and G.P., Writing - original draft preparation: all authors.

*Competing interests.* The authors declare that they have no conflict of interest.

*Acknowledgements.* The work of S.L., V.R. and G.P. was supported by the LabEx Tec21 Investissements d'avenir - agreement n°ANR-11-LABX-0030. F.F. and F.L. were supported by the Labex OSUG@2020 (Investissements d'Avenir, grant agreement ANR-10-LABX-0056).

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
