# Peer review of "Debris-flow surges of a very active alpine torrent: a field database"

_EGUsphere, 2022_

## Author Comment (AC2)

Lapillone et al. report of results from a debris-flow monitoring station in the Real torrent and suggest a data processing protocol for a more consistent and transparent derivation of debris-flow parameters from field observations. I think this is a well-written and well-structured contribution that will be very valuable for the community.

I have only minor to moderate comments and suggestions:

We would like to greatly thank you for your helpful insights and constructive suggestions to our paper. We believe the manuscript to be improved by your clear and thorough feedback, and feel lucky to benefit from your expert feedback.

L 1: "…debris flows" (plural)

Thank you, done.

L 5: "…at the surge scale" (instead of "at surge scale")

Thank you, done.

L 17: I think you refer here to the Nagl et al. (2022) paper (impact forces).

Thank you, correct, done.

L 16: I think one should write "debris-flow monitoring", but "debris flow". Check throughout the manuscript to be consistent.

Thank you, done.

L 100-101: this sentence is unclear. Please re-formulate.

Thank you, done. Reformulated to 'Immature debris-flow surge can also trigger instantaneously high seismic signal, but differ from mature debris flow because the signal consistently drops to zero during the event. This is why the criterion on determination between debris flows and immature debris flows cannot only be based on instantaneously high seismic signals.'

L 145ff: it is not clear to me, how these hypotheses will be tested. Probably it would be better to term them "assumptions".

Thank you, done.

Table 1: I recommend to add a column with the location along the channel (or distance between stations) and a column with the mean slope of the channel reach where stations are located. Both information may help with interpretations given later.

Thank you, done.

Figure 4: for the reader's convenience, I suggest to modify this figure for better readability: (1) the labeling is not intuitive and not consistent with labeling in Figure 5. E.g., why is flow stage termed "rad"? What is geo_21 and geo_21ref? (2) change line color/style to allow an easier differentiation between seismic sensors and flow state. You may also consider to plot diagrams above each other (makes them wider and probably easier to read) or show only one.

Thank you, done. 1)The labelling will be change to be clearer and more consistent between Fig.4 and 5 2) Thank you for this suggestion, that will be much easier to read.

L 169: what is meant with "the least noisy flow stage signal is chosen"? Please re-formulate.

Thank you, done. Reformulated to 'If multiple flow stage signals are available, the most reliable one is chosen, i.e. the flow height sensor that does not present any artefact (unphysical values, very noisy signal, …). Choosing consistently the same sensor across all events when it did not have any malfunctions is preferable."

L 172: bracket is missing.

Thank you, done.

L 204: Delete ""Finally,"

Thank you, done.

Section 3.2: it is not clear to me which type of base level change was used (see L 145ff). Did you compare different assumptions? Are the differences small compare to e.g. peak flow?

True, for stations S2 and S3, only the logarithmic assumption is shown because we assume it to be the most realistic assumption of the three. All the values are recorded in the database and could be shown as uncertainty brackets on Fig. 7. Because it only concerns a few events in our application, we will modify the figure 6 to show these ranges. On Fig.7 we decided not to show the ranges because it will overcrowd the figure in our opinion.

We realized we forgot to clarify something on station S1, which will be added to the manuscript : in our section there are two assumptions for the cross section shape (which are described by Bel in [1]). An average is shown as we cannot assume preference between the shapes easily, and the range of variability is low (see Fig. 11, points in blue). This will be clarified in the revised manuscript.

L 210: write "literature", not "litterature" throughout the manuscript.

Thank you, done.

L 212: I recommend to stay more general and replace "viscosity" with "mixture composition", that's safer.

Thank you, done.

L 234: unclear sentence. Please re-formulate.

Thank you, done. Reformulated to "It seems that debris flow surges of large volume require a strong inertial input to flow, as there are no subcritical Froude numbers for volumes of the selected range. Their heavy granular content, increasing their macroscopic viscosity,cause that subcritical, slower flows, with high volumes would stop or deconstruct."

L 248: unclear. What do you mean by "witnesses".

We meant proxy, this sentence has been changed to' Figure 7c shows no definitive relationship between proxies of inertial and potential inputs in the flow.'

Section 4.3: I encourage to add some interpretation of the observations and measurements with regard to channel slope at the stations and distance between stations. E.g., is deposition to be expected at flatter reaches between stations?

Thank you for the suggestion. Local slopes will be added to Table 1. The slopes measured show that these are transfer zones where both erosion and deposition are observed, which is more thoroughly investigated in [2]. Interpretations will be added in text.

Figure 10: I am wondering other symbols color may help to make the time component more readable. Probably grey-scale increasing over time?

Thank you, done.

Figure 11: can you add the range of uncertainty from volume and peak discharge estimates (assumptions on base level changes, L 145ff) to the diagram? In the Figure caption you may write "debris-flow surge volume".

I am not sure I understand weather you mean adding the uncertainty for each assumption ? If so, adding them makes the graph very hard to read. The three symbols represent the three assumptions.

Thank you, done for the second part.

We thank you for your time spent in helping us improve the work thanks to your constructive comments.

References :

[1] : C. Bel, Analysis of debris-flow occurrence in active catchments of the French Alps using monitoring stations, Ph.D. thesis, Université Grenoble Alpes, 2017

[2] : J.I. Theule, F. Liébault, D. Laigle, A. Loye, M. Jaboyedoff,Channel scour and fill by debris flows and bedload transport, Geomorphology,Volume 243, 2015,

---

## Author Comment (AC3)

This study presents the results of 10 years of debris flow monitoring in Real catchment, SE France. Presented data are original and novel and have potential to be further exploited, especially in DF modelling studies where physical parameters need to be defined. The authors mention that they present a 'protocol' to analyze DF but I don't see that clearly in present structure of the study (or what do you actually mean by 'protocol'? is it a methodology how you prepare Table S1? maybe a flowchart figure could help to understand that; please clarify or consider re-framing). Apart from that framing issue, I recommend some additions (see details below) and I have a couple of comments:

Thank you for the very precise and helpful review. We considered this a protocol as it is a method for post processing raw monitoring data, but we acknowledge how this can be misleading. This will be corrected to correspond more to accurate vocabulary. The aim of this submission is to present a methodology for peer review, the provided data are an application of this method and are here for illustration and to show the potential of using this method on multiple station. The introduction will be slightly modified to make this framing more clear to the reader. Thank you for your constructive comments on this paper both on precise sections and on this framing issue.

L28-30: apart from scientific publications, I wonder whether the DRR authorities or other authorities in charge of DF management / monitoring collect and could provide more data?

There are no monitoring stations managed by authorities in France as of today. There are some detection devices installed mainly to manage road safety problems (traffic lights triggered by sensors). However, some projects are being developed (very early planning), so we think now is a good time to agree on a unifying peer-reviewed methodology.

L52-54: does this open access database already exist? Is it planned to be created? How would you motivate people to contribute their data?

In addition to the data provided in the appendix of this paper, the idea is to make the database completely available online, including raw data from research team willing to share them. We are currently opening a dedicated repository on an online open repository. We will most likely provide a DOI link toward this first sample of the dataset in the next versions of the paper. Meanwhile, this paper will help to disseminate and share the idea one how to identify debris flow surges. We will present it in the next conferences and looking for collaborators to feed this database. The aim of this current paper is to have a first peer-review of the methodology so that we can apply it to more sites and then jointly publish analyses that would not be site specific and would have a broader interest because based on the bigger dataset .

L57: in a catchment

Thank you, done.

L62: is this your 'protocol'? if so, please name it accordingly (please also consider visualization of individual steps in a flowchart figure; see above)

Section 2.1 will be renamed correctly. We see the advantage of a flow chart and initially decided to skip one due to an overcrowding of figures in the paper, but we will add one or modify Fig.1 for clarification in this section.

L89: please provide a reference to this hypothesis

Thank you, described in [1]. Reference has been added in the paper, thank you.

L89: Focusing on data processing

Thank you, done.

L100-103: is this seismic signal analysis something you actually used and presented in the results of your study? If not, it should be removed from Materials and Methods section

Geophones record the vertical component of the seismic signal. We are using this vertical component of the seismic signal for sections 2.1.2 and 2.1.3. The signal is a proxy of a PSD made analogically (all frequencies are combined as one value). We agree that this might be misleading and it will be clarified in text.

Fig. 2: please clarify what is the use of geophone data in your study? How does it contribute to summarizing Table S1

See section 2.1.2 and 2.1.3 where the geophone signal is used to determine surge extent and velocity.

L141-150: I don't understand what you mean here; if you aim at presenting widely-applicable methodology (protocol), you should be as instructive as possible

We agree with you that this needs to be as instructive as possible. This will be reformulated accordingly, for example : "Accounting for the variability in channel is necessary (e.g. width, bed level, shape). Due to the debris flow event, scouring or filling can occur both vertically and horizontally to the cross-section. For each station, assumptions on cross section shape have to be made, and questions about variability in the channel have to be answered. For example, assumptions on cross section shape and change must answer to whether the channel can be scoured/ filled in that section and whether there is a difference in the preferred channel between low and high flows. Assumptions have to be as precise as possible using the information on the channel at this point (e.g. local obstructions to the flow are known, non erodible banks, ...)."

Fig. 3: is dotted red line for max (isn't it rather form min) and vice versa?

Max hypothesis means hypothesis which maximalizes the sediment volume transported, and maximalizes the effective height, which corresponds to the dotted red line. This will be clarified in text.

L151: please consider separate 'study area' section with more details on general physical geographical setting

A more thorough description of the study site will be added as well as reframing this section more precisely. The complete description of the site can be found in Hürlimann et al [2] and Bel [3].

Fig. 5: please consider adding information about elevation (basic contour lines)

Thank you, done.

L178-179: not clear, please clarify what you did at this step?

For one surge, the cross correlation coefficient was not satisfactory. The visual method consists in taking 4 points : two on the first geophone signal : before and after the first front; and two on the second geophone signal : before and after the first front. This can also be done on the flow peak depending on flow shape. Figure S3 in supplementary data presents this specific case. We are going to reformulate this section to clarify.

L181: I don't understand point (ii) -in Table S1, you present rainfall data with precision to 1 decimal place

Table S1 only presents maximal values and cumulated values, but temporal signals are saved in the database. They are of no use for this study so they are not presented any further but will be present in the database.

L184: I suggest to consider re-naming this section (e.g. observed DF, or similar)

Thank you, done.

L185: how do you defined 'significant' evet? Is this where the seismic signal comes into play? Please clarify

This is an arbitrary decision. We decided from knowledge on the types of flows we record on these stations. This will be clarified in the text.

L190: how do you know there was natural variability if the measurements didn't work?

The flow stage sensors did not work correctly but the geophone signals did, and they showed only debris flood and debris flow activity during this time.

L202: please unify Froude numbers to L194

Thank you, done.

Fig. 6: does it make sense to plot measurements from three monitoring stations in one curve? Considering erosion / depositional processes on a way, I suggest plotting separate curves for individual monitoring stations

The goal of this work and this figure is not to analyze site specific events on this precise torrent, but rather to provide ranges for natural debris flows, which are often not available in the literature. Rather than making multiple figures, we found more striking to show that the three stations have similar ranges.

Fig. 7: what is the reason for plotting these values? Would you expect correlation or causality? I suggest you to quantify possible correlations.

Due to mixture composition and difference in flow hydrographs shape observed in debris flows (e.g. see [4]), the expectation is that there would be a variable but we would not know at which range. In

section 4.1, different interpretations are explained and the links between variables is explored. The expectations in term of causality and correlation will be more clearly added to section 3.2.

The dataset is too small to explore properly correlation between variables in the statistical sense, but this would be of interest once collaborations allow to have more data.

L212: lack of trend or no correlation?

Thank you, done.

L218: this value is beyond what is shown in Fig 7a (max 2 000 m3/km2); please check

Thank you, done. One datapoint was not shown indeed.

L243: viscosity varies

Thank you, done.

L245-247: this is not clear to me, please reword this sentence

This parts deals with the variability of flow hydrograph. Reworded to "Debris flows have a very variable flow hydrographs [4, among others] due to a wide range of flow mixture. This leads to similar volumes of debris flow surges to be caused by different types of flow hydrographs : shallow surge which last for a long duration or very intense, high, but short surges.".

L256: there is no part b-c in Fig. 1

Sorry, Fig.5, now corrected.

L259-271: I'm not sure I get what you want to say here

If we intend to perform sediment balance studies, a measure of the bed-load transport is necessary. The current monitoring and the data presented in this paper alone does not allow for the sediment cascade to be studied. Figure 8 shows a cycle of sediment scour and filling in the channel which is a proof that the sediment activity is high in this catchment. However, such cycles cannot be detected in Figure 9. If it were possible to see this geomorphic cycle, the cumulated volumes of the surges passing at S1 would be found to be equal to the cumulated volume at S2 over the years. Any of the deposit at S1 or between S1 and S2 would then be exported downstream. To summarize, debris flows alone are not sufficient for sediment balance investigation in such catchment and bed load has a significant impact (which is consistent with [5]).

L272: insights from multitemporal high-resolution images might help to answer some of the remaining questions raised in this section (e.g., a remobilization of material deposited by previous event(s))

Agreed, but no such multitemporal high resolution images are available on this period.

Thanks for the comment, added to the text.

L297-298: see above

Correct.

L312: ranges of what?

Thank you, changed to Analysis of the ranges of the physical characteristics of the events.

L321: what do you mean by 'proof of concept for data processing?'; were there any doubts about it?

Changed to conceptualization of a widely applicable protocol for debris flow data processing.

L323: in the paper, you don't say much about how this collaboration and common database should like

*More information on this will be added. This paper is meant to begin the collaboration by having a protocol every station can use. see reply to L52!*

L329: your data are site-specific rather than representative

We completely agree. This is why we want to have a collaborative database. We are purposefully waiting for the collaborations to be in place to publish the database. This paper intends to publish the methodology and the illustration on a specific site so that other sites can apply a methodology that has been peer reviewed.

- - -

To sum up, some interesting field data are presented and I recommend acceptance of this study as soon as some moderate revisions are made.

We would like to thank you for the time spent on this very helpful and valuable review. The comments made will help our paper to be more clear and straightforward to read.

References :
[1] Hungr, O. (2005) Classification and Terminology. In: Jakob, M. and Hungr, O., Eds., Debris Flow Hazards and Related Phenomena, Springer Verlag, Heidelberg.
https://doi.org/10.1007/3-540-27129-5_2
[2]Marcel Hürlimann, Velio Coviello, Coraline Bel, Xiaojun Guo, Matteo Berti, Christoph Graf, Johannes Hübl, Shusuke Miyata, Joel B. Smith, Hsiao-Yuan Yin, Debris-flow monitoring and warning: Review and examples, Earth-Science Reviews, Volume 199 ,2019,
https://doi.org/10.1016/j.earscirev.2019.102981
[3]  C. Bel, Analysis of debris-flow occurrence in active catchments of the French Alps using monitoring stations, Ph.D. thesis, Université
Grenoble Alpes, 2017
[4] Mitchell, A., Zubrycky, S., McDougall, S., Aaron, J., Jacquemart, M., Hübl, J., Kaitna, R. and Graf, C., Nat. Hazards Earth Syst. Sci. **22**(5), 1627–1654 (2022).
[5]  J.I. Theule, F. Liébault, D. Laigle, A. Loye, M. Jaboyedoff,Channel scour and fill by debris flows and bedload transport, Geomorphology,Volume 243, 2015,

---

## Author Comment (AC4)

**General comments**:

The manuscript by Lapillonne et al. addresses an important concern with a proposal for a uniform protocol for event analysis at gaging (automated monitoring) stations and represents a valuable contribution to the debris-flow community with the information provided in an event database from the French Réal monitoring stations. The authors show how the procedure can be applied to their own measurement series. They critically examine their own measurement data (quality, acquisition methods, positioning, etc.) and show the possibilities but also the limitations of the proposed methodology. The work also shows that additional information (metadata) and event descriptions can be very important to interpret measurement data correctly. The manuscript is basically well written in a good and understandable English and the procedure is described in a comprehensible way. It should therefore be of great interest to the community and fits well into egusphere. Nevertheless, various statements are not formulated precisely enough and require more detailed explanation or additions.

W are very grateful for your interesting and constructive review. Your insights are very valuable and the manuscript is improved by your suggestions and expert comments.

**Specific Comments**:

12: what means "precise"? Can the authors specify how precise is precise enough... perhaps by giving a measure of uncertainty?

Thank you for the comment. In this case we mean that complete debris flow measurements, including flow height sensors, imagery, detection devices, … are hard to install due to the harsh condition. This will be clarified as the use of the term 'precise' is not conveying the message correctly.

12: In general, please check how debris flow is written when used "adjectivally", i.e. in combination with other nouns. Either in this case always write a hyphen between debris and flow (debris-flow), e.g. debris-flow measurements or never use a hyphen. Both variants exist and personally, I prefer the hyphenated style.

Thank you, corrected!

15: use "observed" or "measured" instead of "monitored".

Thank you, corrected!

51: The authors need to better derive why the characteristic values of individual surges are so important and differentiate between characteristic values of an entire debris-flow event vs. individual pulses or surges, which can sometimes differ greatly within an event.

Characteristics of surges are the focus of this paper and protocol because the numerical models getting interested in the physics of debris flow and impact studies focus on the surges. We will better explain this difference.

53: write "automated debris-flow monitoring station"

Thank you, corrected!

57: Not all can realize instrumenting multiple sites in a catchment area. This depends on the one hand on the location and on the other hand on the financial resources and manpower. Please specify that you mean the measuring points along a channel covering different channel sections and not stations at several torrents from sub-catchments. Perhaps one could introduce to differentiate that the entire measuring infrastructure in an area is called automatic debris-flow measuring station and the individual stations are called sub-stations or similar.

Thank you, that is a great observation. Vocabulary has been corrected but we would like to point out that having multiple sub stations is not in any way a requirement for this protocol. We just took advantage of our 'luck' concerning our instrumentation in this paper.

Fig. 1: Please complete the associated (or: derived) surge parameters also in the figure 1

Thank you, corrected!

65: It is not clear where the value comes from and how it is justified. If this is a minimum requirement, then conversely it should say less than or equal to 2 Hz. The changes in flow height can be very rapid and if the resolution is too low, you may miss the peak value. There is also a larger discrepancy between the different measurement methods used, including time delays in determining the value.

You are correct, there is an inconsistency here. 2Hz was initally chosen because of the refresh ability of the ultrasound sensors of our setup at that time, as well as the cut off frequency of our acquisition system. If you are interested in having more information on these sensors, a publication is currently being written on the sensors used in this station.

  This has been modified to "flow stage measurements with representative frequency sufficient to describe accurately the flow front rise on the hydrograph".

66: It is essential to know the value before, during and after the passage of a surge as accurately as possible to minimize calculation errors.

Thank you, corrected! This has been added to the requirements for clarity.

67: please write "mean velocity" because here an average value of the propagation of the surge is calculated and not the instantaneous and local velocity

Thank you, corrected!

71: This vague statement about sufficiently close locations must be specified and substantiated (e.g. erosion, deposition, alteration of flow path, etc.)

Thank you, we will add a sentence to clarify what we mean by sufficiently.

"These measurements must be done at sufficiently close locations to reasonably assume that the measured flow stage is associ-
ated with the measured surge velocity. Between two sensors, there should be no major change in flow path, channel width and slope to ensure that the geomorphological processes are consistent along the interdistance."

81: write: "…is the maxiumum volume of the flow depth $h$ [m]

Something was wrong in the sentence indeed, it has been corrected to is the maximum value of the flow depth $h$ [m]!

83: provide a reference

Thank you, done, a reference to [2] will be added here.

95: use "flow height" instead of flow stage.

Thank you, corrected!

98: not clear, if raw data is used. Please specify. There are also alternative methods in which pulses are counted that exceed a specified output signal threshold (see: e.g. Abancó et al., 2012 - https://doi.org/10.3390/s120404870)

Thank you, raw data was used. The text will be corrected accordingly and the alternative method will be mentioned.

104: It is not made clear whether an and-link of flow height (please be consistent and use "flow height") and geophone signal or an or-link is used to divide the debris-flow event into surges, and exactly when the separation occurs. Please specify more precisely.

Flow height has been correctly rewritten in the figures and in the text. An and-link is used. This part is reformulated to clarify.

119: see comment for line 71

This distance will be more precisely specified, similarly to our answer to L71.

120: How is the distance determined? Is the exact flow path measured (possibly a variable value) or is the direct visual distance meant? Please specify more precisely.

The distance is taken as the average flow path between the sensors i.e. the path of the main channel between the two sensors. This assumption of it corresponding to the flow path is not very precise indeed. This has been added in text.

134: Please note that debris flows often have a convex curved surface in the transverse profile. This must be taken into account when determining the flow height and the characteristic values derived from it (see: e.g. Jacquemart et al., 2017 - https://doi.org/10.1007/s11069-017-2993-1)

Yes, this has to be a case-by-case assumption, depending on the location of the debris flow. In our case, this was covered in Bel, 2017. This point will be added to the text.

145: please mention also Fig. 11 here

Thank you, corrected!

162: More out of curiosity: what is the advantage of placing a geophone downstream the flow height (stay consistent) sensor?

This is mainly to ensure that we have two zones to compute the velocity as the sensors are in a difficult environment they can fail relatively often, so multiplying the measurements make it safer.

168: The selection procedure should be described in more detail, e.g. by stating decision criteria and their weighting. Otherwise, the data reduction remains not transparent for others and could be applied incorrectly.

The selection is mainly based on a visual estimation of which sensor worked better : usually the sensor giving the highest values are chosen to avoid having a noise on the sensor that is of the same order of magnitude as the measured values. If one of the sensors failed (oddly high or oddly low values temporarily) then we will prefer the others. The influence of this choice has been tested and remains marginal.

174: What is meant by "...a clear appearance of the debris-flow behaviour...". Please specify more precisely.

This is in reference to section 2.1.2. We will change the text to make this point more clear for the reader.

181: Rainfall measurement is important but not of interest in this paper. This information could be omitted.

Thank you, we initially mentioned it for future user of our data, but this could be moved to supplemental information!

185: What are "significant events"? Please specify more precisely.

This is arbitrarily defined using the expertise on the station. Added to text.

Fig. 7a/b/c: For better readability and to avoid confusion, use a different color code for each legend on the right side. In the legend, a space is missing after c) and the word "literature" is not spelled with a double t (also occurs several times in the continuous text).

Thank you, corrected!

206: Turn over sentence to: "The surge volume was normalized by the catchment area to cross-compare measurements performed at different stations, but also to help transferring these results to other catchments."

Thank you, corrected!

210: literature

Thank you, corrected!

212: add "clear trend"

Thank you, corrected!

213: use "carful" instead of "prudent"

Thank you, corrected!

220: Please check whether the calculation for the Illgraben data uses the total catchment area (11.69 km2) or only the area of the active contributing sub-catchment (4.83 km2). The large difference could also be due to this. cf. Hürlimann et al., 2019 , p. 12

In this calculation, we used the total catchment area. The contributing area would not be the only contributor to the total volume of the flow. Thank you for the comment, we will add this precision to the text.

226: literature

Thank you, corrected!

237: please explain what your mean by "heavy surge". Also not clear what you mean by "with the topography of this catchment…". Please specify more precisely.

High in volume, that is corrected in the paper now. The reference to topography was unnecessary : high volumes are not as present because the hydrology of the catchment allows for sediment transfers to occur rather often (see Bel [1]) and the steep slopes lead to low yield criterion of the accumulated sediments.

243: what is the driver to flowing conditions then? Please specify more precisely.

Composition of the mixture changes the mobility of surges. We will specify more precisely in text.

255: Please provide an additional figure or a supplement to an existing figure with a longitudinal profile showing the position of stations S1, S2 and S3. Moreover, just out of curiosity, is the area regularly flown (with a drone) to calculate a sediment budget. Such data could provide valuable additional information.

The additional figure will be added to supplemental material. Unfortunately, we have not had the opportunity to regularly calculate sediment budget on this catchment. Such data would be very precious to our study and could be something to investigate in the future.

Fig. 8: Write (pictures from the authors). Add "note persons in pictures c) and d) for scale"

Thank you, corrected! Persons are now also lightly highlighted to better detect them in gray scale.

Fig. 9: If applicable, make the figure wider (extend the timeline) and connect related events with a line.

Thank you, corrected!

273: call them "sub-stations"

Thank you, corrected!

283: Here it becomes clear, among other things, why additional information and event descriptions are of great importance as a supplement to the pure measurement data at automatic monitoring stations.

Fully agree, this will be added to the text.

Fig 10: To make the figures easier to read, I have connected the points that belong together with a line. Could possibly be supplemented. In addition, the symbols could, for example, be displayed larger or smaller depending on the volume, which could allow additional statements to be made.

Thank you, corrected for the line, and grey scale by date was added, thanks to other comments. However, we think that changing the size of the symbols would make the graphs harder to read and might not be in our best interest. We will try and see if we can implement this in a useful and clear way.

Fig. 11: Note that the data basis of the empirical datasets are not based on measurements of individual surges (at the very most on the dominant surge of an entire debris-flow event)!

Thank you, this was more clearly added to the 4.4 section and in the title of the figure.

322: As was mentioned, data sets already exist. Accordingly, the contribution represents another and not the first data set, but it is based on the evaluation of surges. This still needs to be specified more precisely.

The wording was misleading, we meant a first dataset 1/for the Réal torrent & 2/ with this protocol for surge evaluation. This was corrected, thank you!

We thank you for the time spent on this very helpful review. The work will benefit greatly from your insights.

[1] C. Bel, Analysis of debris-flow occurrence in active catchments of the French Alps using monitoring stations, Ph.D. thesis, Université Grenoble Alpes, 2017

[2] Hungr, O. (2005) Classification and Terminology. In: Jakob, M. and Hungr, O., Eds., Debris Flow Hazards and Related Phenomena, Springer Verlag, Heidelberg. https://doi.org/10.1007/3-540-27129-5_2